# Mast Cells in Regeneration of the Skin in Burn Wound with Special Emphasis on Molecular Hydrogen Effect

**DOI:** 10.3390/ph16030348

**Published:** 2023-02-24

**Authors:** Dmitri Atiakshin, Mariya Soboleva, Dmitry Nikityuk, Nataliya Alexeeva, Svetlana Klochkova, Andrey Kostin, Viktoriya Shishkina, Igor Buchwalow, Markus Tiemann

**Affiliations:** 1Research and Educational Resource Center for Immunophenotyping, Digital Spatial Profiling and Ultrastructural Analysis Innovative Technologies, Peoples’ Friendship University of Russia, 117198 Moscow, Russia; 2Burdenko Voronezh State Medical University, 394036 Voronezh, Russia; 3Federal Research Centre of Biotechnology and Food Safety, 109240 Moscow, Russia; 4Institute for Hematopathology, 22547 Hamburg, Germany

**Keywords:** burn wound, skin, fibrous extracellular matrix, mast cells, molecular hydrogen, regeneration

## Abstract

The mechanisms of regeneration for the fibrous component of the connective tissue of the dermis are still insufficiently studied. The aim of this study was to evaluate the effectiveness of the use of molecular hydrogen on the local therapy of a II degree burn wound with the intensification of collagen fibrillogenesis in the skin. We analyzed the involvement of mast cells (MCs) in the regeneration of the collagen fibers of the connective tissue using water with a high content of molecular hydrogen and in a therapeutic ointment for the cell wounds. Thermal burns led to an increase in the skin MC population, accompanied by a systemic rearrangement of the extracellular matrix. The use of molecular hydrogen for the treatment of burn wounds stimulated the regeneration processes by activating the formation of the fibrous component of the dermis, accelerating wound healing. Thus, the intensification of collagen fibrillogenesis was comparable to the effects of a therapeutic ointment. The remodeling of the extracellular matrix correlated with a decrease in the area of damaged skin. Skin regeneration induced by the activation of the secretory activity of MCs may be one of the possible points of implementation of the biological effects of molecular hydrogen in the treatment of burn wounds. Thus, the positive effects of molecular hydrogen on skin repair can be used in clinical practice to increase the effectiveness of therapy after thermal exposure.

## 1. Introduction

The complete restoration of the skin as an organ appears to be one of the crucial issues in regenerative medicine nowadays. The morphological aspects of the wound process are the focus of attention for various researchers. Burn wounds are of special significance compared to other skin injuries due to both the specificity of the wound process and the long-term restoration of skin integrity [1,2]. The restoration of the skin as an organ after thermal alteration is directly related to the state of the extracellular matrix of the dermis, including the amphora and fibrous components. The extracellular matrix, in addition to the structural components that ensure the integrity of the skin, includes a wide range of informative factors, such as chemokines, cytokines, growth factors and other molecules with a signal function that determine the morphogenesis of a specific tissue microenvironment during the development of adaptive reactions and pathological conditions [3]. The connective tissue and the structural and physiological framework of the skin dermis is determined by the ratio of the fibrous component, mainly consisting of fibers with type I and III collagens [4]. Mast cells (MCs) have an essential role in the formation of the connective tissue base of the skin. They are able to influence the state of the extracellular matrix, having important sensory and effector properties [5,6]. MCs possess a number of important functions for providing the molecular mechanisms of skin wound regeneration [7,8,9,10].

Activated MCs control the key events in all the phases of wound healing—inflammation, proliferation and remodeling—using their receptor and effector properties [10,11,12]. Specific proteases are of great significance in the realization of the biological effects of mast cells for the restoration of the damaged skin structures [13,14,15,16]. The positive effects of molecular hydrogen on the rate of wound healing are well known [17], but the mechanisms of the discovered effects of molecular hydrogen remain unclear. There is a fundamental potential and numerous factual materials on the histotopographic polymorphism of the MC fiber-forming function, depending on the state of the specific tissue microenvironment [15]; however, their significance in the implementation of collagen fibrillogenesis under various protocols of burn wound management, including using molecular hydrogen, remains to be elucidated. The aim of the study was to evaluate the effectiveness of the use of molecular hydrogen on skin regeneration processes associated with the activity of collagen fibrillogenesis and the activity of mast cells in the treatment of skin burn injuries.

## 2. Results

### 2.1. Spontaneous Healing of the Skin after a Burn

Three days after thermal exposure, spontaneous healing was characterized by an increased participation of MCs in the regulation of local homeostasis compared to the morphofunctional criteria of the intact skin. MCs were not practically detected on the burn surface, which could be related directly to the action of the thermal factor and an increased consumption of secretory products during the thermal damage, which leveled metachromasia. MCs were properly detected either in the lower dermis, near the hypodermis, or in areas surrounding the wound surface along the periphery, where they actively excreted granules and mediators into the extracellular matrix (Figure 1). Active degranulation was accompanied by certain loss of the granules’ metachromaticity, which evidenced the active use of heparin in the processes of post-wound morphogenesis. The number of granules in the MCs decreased (Figure 1A). Attention was drawn to the fact that there was limited MC participation in the process of fibrillogenesis, with morphological signs of secretion (Figure 1B–E). It was possible to occasionally detect signs of a loose network of thin fibers in the area of secretion (Table 1 and Figure 1F–J).

At 7 days, MC migration led to their location in the peri-alterative areas of the skin dermis, including the direct area under the burn surface (Figure 1M). MCs were located between bundles of collagen fibers. Some of them were large fragments of the cytoplasm or non-nuclear parts of the cell. The strategic location of the nearby elements of the vascular bed, including the microcirculatory region, allowed for the more intense regulatory effects to ensure local homeostasis. Compared with the previous period, it was possible to report the activation of fibrillogenesis, which was manifested by the formation of thin collagen fibers in the pericellular region and MC location near the stromal components of the microvasculature (Table 1 and Figure 1K,L,N).

At 14 days, MC involvement in the remodeling of the extracellular matrix became more expressed. The greatest migration of MCs was noted in the peri-wound areas of the skin compared to earlier healing periods. MCs sometimes formed large clusters over a large area, taking an active part in the direct or indirect regulation of the skin dermal remodeling. Histotopographically, MCs were localized along the periphery of the burn zone (Figure 2A); however, some of them were located in the regeneration zone under the burn surface (Figure 2B). Concurrently, MC participation in the formation of collagen fibers markedly increased; this occurred in close functional collaboration with fibroblasts and fibrocytes (Table 1 and Figure 2C–F).

Patterns of small MC group formation were observed more often, providing evidence for an increased intrapopulation cooperation to enhance the mutual effects of the extracellular matrix formation (Figure 2D). MCs that were incorporated into bundles of collagen fibers continued their secretory activity (Figure 2G). Active degranulation led to a gradual depletion of the MC mediators and was accompanied by a decrease in the metachromatic properties of the secretome (Figure 2H). The processes of fibrillogenesis were also observed in the MCs remote from the wound surface at considerable distances. (Figure 2I,J).

### 2.2. Fibrillogenesis of Collagen under Therapeutic Ointment Application

At 3 days after burning, ointment application resulted in greater MC preservation in the structures of the skin burn surface compared to that of the animals in the spontaneous healing group. There was an impression that the MCs were located in the lacunae, in which case MC secretory activity was maintained at a certain level (Figure 3A–G).

The number of mast cells in the skin, including that at the moment of exposure to alternative factors, should be considered as the degree of the organ resistance to external agents.

Therefore, MC preservation in the thickness of the dermis under ointment application allows for the maintaining of higher levels of regulatory molecules and biologically active substances in the damaged area, which, among other things, are necessary to ensure better trophism of the damaged structures and to protect the preserved structures from the impact of other potentially pathogenic components of a specific tissue microenvironment, including free radicals. In addition, it is possible that the therapeutic ointment application provided better conditions for MC migration, allowing them to move in the thickness of the dermis and take strategically more beneficial positions for local homeostasis coordination. Yet, MC detection in lacunae with walls containing a high amount of fibrous structures may indicate an attempt to preserve the MCs from surrounding tissues in order to maintain viability under adverse conditions, including hypoxia.

MCs had different levels of fiber formation, from low (Figure 3G) to high (Figure 3C,H,I), in the preserved areas of the skin dermis. However, on average, the relative content of thin pericellular impregnated structures increased significantly (Table 1).

At 7 days of wound healing with ointment application, MC participation in the fiber formation increased markedly compared to the previous period of observation (Table 1 and Figure 3J,K). Sometimes, there was an impression of fibrous matrix formation at the site of an intense MC degranulation (Figure 3K). MCs were detected more frequently around the microvascular vessels (Figure 3J,K). This increased the MC secretory activity in the peri-venular space, both in the direction of the basement membrane of the endothelium (Figure 3J) and the fibrous skeleton outside (Figure 3K).

At 14 days, the process of fibrous component formation actively continued. This activation was comparable to the processes observed in the spontaneous healing group (Table 1). Similarly, most MCs accumulated along the periphery of the wound skin area. Histotopographically, a greater number of mast cells were detected directly under the thermal damage zone. Mast cells formed small groups of 2–3 cells, which sometimes directly contacted each other (Figure 4A,B,G) or were in the zone of paracrine effects (Figure 5 and Figure 6). Sometimes, MCs acted as a sort of connection center for rather large bundles of collagen fibers (Figure 4A,C,K). MCs continued to accompany the microvasculature (Figure 4B) and to actively degranulate (Figure 4C,F,G). It was noteworthy that histotopographic differences were formed in the activity of fibrillogenesis over 14 days. There were identified zones with a completed remodeling of the extracellular matrix; they were characterized by the dense integration of MCs into bundles of collagen fibers. In addition, the spatial remodeling of the fibrous skeleton of the damaged dermal part continued; MCs there occupied an intermediate position between large bundles of collagen fibers. Concurrently, MCs in some loci retained a high potential to initiate the assembly of collagen fibrils, most often at sites with a lack of fiber bundles in the extracellular matrix.

### 2.3. Molecular Hydrogen Application in the Burn Management

At 3 days after burning, there was an impression of a better MC preservation in the loci of the damaged skin compared to the indices of the animals from the spontaneous healing group (Figure 5). Most often, they were actually hidden in a damaged fibrous skeleton, wherein the granular structure and the severity of metachromasia were lost (Figure 5A–C).

It should be noted that, starting from a distance of more than 50 μm, MCs were characterized by a high activity of biogenesis on the fibrous collagen structures in the skin dermis in the peri-burn areas (Table 1 and Figure 5). MCs had an intense degree of secretion. Large MCs, which were localized in the peri-burn surface, often contacted smaller satellite fragments of the cytoplasm filled with granules (Figure 5D,F,G), and reticular fibers were determined between them. The vast majority of the isolated MCs were accompanied by signs of the initiation of thin collagen fiber formation (Figure 5D–F).

At 7 days, MC groups were detected under the burn surface. Most likely, this was a directed migration of MCs in the newly formed structures of the skin dermis. Concurrently, MCs were characterized by active secretion (Figure 5G). Greater amounts of MCs, if compared with the animals of the spontaneous healing group, continued to be detected in the damaged dermis, albeit with signs of partial homogenization of the secretory material (Figure 5H,I). Mast cells associated with fibroblasts were actively involved in fibrillogenesis (Figure 5J). Notably, the number of MCs was comparable to that in animals from the experimental group, where the ointment base was applied to the damaged area. Interestingly, if compared with the therapeutic ointment effects, MCs under molecular hydrogen application often formed; in terms of histotopography, migration groups were located along a certain trajectory in the connective tissue base of the skin (Figure 5G).

At 14 days after burning in the animals of the group where the wound was irrigated with water enriched with molecular hydrogen, there was detected the highest fibril-forming activity of MCs compared to the other experimental groups (Table 1). This occurred both in the sub-burn area and around the wound, at a considerable distance from the injury site. MCs often formed clusters that were actively secreted by expelling individual granules outside the cell (Figure 6A,B). At some loci, secreted granules were selectively positioned in the extracellular matrix to form a prototype of the future fibrous structures of the extracellular matrix (Figure 6A,B,D,H). Non-nuclear regions of the MC cytoplasm were frequently detected, which, if considered by morphological features, could serve as a locus of initiation of the reticular fiber formation (Figure 6E,G).

Notably, the degree of metachromasia in the secretory granules gradually decreased, evidencing an active entry of heparin into the extracellular matrix, with an inductive change in the integrative metabolic buffer environment (Figure 6G). Groups of secretory granules, which could be located between bundles of collagen fibers, were detected in the dermis of the skin.

## 3. Discussion

The MC involvement in the extracellular matrix formation and the coordination of the remodeling mechanisms during spontaneous wound healing increased from the 3rd day of the experiment to the 14th day of observation. This was achieved by increasing the intra-organic number of MCs in the skin dermis and their activity in the fiber formation. The remodeling process can be considered as a systemic reaction using the regulatory properties of MCs not only in the area of damage, but also in the surrounding areas of the normal skin. The impact of the ointment had positive effects on the recovery intensity of the fibrous extracellular matrix of the connective tissue after a skin burn to a more active extent, if compared to the spontaneous healing group. It is possible that these effects are associated with better MC preservation and participation in the regulation of the dermal connective tissue base regeneration. The therapeutic effects of molecular hydrogen, as well as that of a therapeutic ointment, were manifested by an improved regeneration in the area of thermal damage and the peri-burn zone with an intensified recovery of the extracellular matrix, mediated by the MC participation in the remodeling of the extracellular matrix of the skin dermis.

Obviously, the intensity of the skin regeneration after a thermal injury depends on the intensity of the burn effect. The study of the biomaterial on the first day of the experiment revealed a significant reduction in the volume of the mast cell population in the burn surface of the skin, with a depletion of the pool of both small cells in the sub-epidermal space and larger cells located below. This phenomenon is definitely associated with both the direct destructive effects of a burn on the MCs and the effects of the protective function implementation under thermal exposure. A massive release of secretome components in response to a damaging effect could lead to a decreased amount of heparin and other secretome glycosaminoglycans, which technically aggravate their detection due to the loss of histochemical properties. It is possible to only highlight a group of animals with the therapeutic ointment application, in which viable MCs could be found in tissue detritus 3 days after burning. Thus, MCs can also be considered as a factor of nonspecific resistance to burn exposure. However, in 3 days and in 7 days, it was possible to detect MCs that had signs of lost functions at the end of the life cycle and in the morphological criteria of preserved functional activity in the damaged areas of the skin under therapeutic effects. Seemingly, both the ointment and molecular hydrogen are able to modulate MC resistance with the preservation of the functional activity. On the other hand, therapeutic effects may be associated with the stimulation of MC migration from undamaged areas of the dermis to the alteration zone. In any case, the therapeutic effects of the ointment and molecular hydrogen led to an enhanced remodeling role of the MCs and an increased rate of the extracellular matrix regeneration, which was expressed, in particular, by the level of initiation of new collagen fiber formation. In addition, the histotopographic features of the MC migration, with accumulation in some loci of the dermis, were noticed under molecular hydrogen application—a fact that could have contributed to the faster isolation of the wound surface from the external environment and a more intensive formation of the fibrous extracellular matrix.

During the wound healing process, MCs take an active part in the regeneration of the structural components of the connective tissue surrounding the wound. In particular, this is entirely obvious when studying the number of MCs in the normal skin and under a burn wound simulation. It is noteworthy that the number of large MCs with a high secretome content was highest mainly at the periphery of the wound. The special intradermal distribution of the MCs in relation to the burn wound appears to be of a certain pattern in terms of histotopography. An increased secretory activity of the MCs in the area of the wound periphery causes a systemic reconstruction of the extracellular matrix, a fact that contributes to the wound healing stimulation resulting from the intensification of the extracellular matrix proteins production; the latter ultimately provides for the elimination of the defect due to the epithelium covering the newly formed dermis material towards the center of the wound defect.

The ongoing therapeutic effect of the secretory activity of MCs deserves special attention. Mast cells are able to have direct and indirect effects on the rate of the fibrous component formation [15]. Influencing the production of collagens by fibroblasts, MCs create the molecular basis for the formation of the fibrous extracellular matrix [15]. In addition, MCs are able to be directly involved in fibrillogenesis, creating the required conditions for procollagen molecule association into supramolecular fibrous structures, including that with the help of heparin [18,19,20]. Numerous facts indicate the MC potential as a target for pharmacological agents, including those obtained with various methods investigating its therapeutic effects on wounds. This is manifested in the degree of degranulation activity and in the activity of the connective tissue extracellular matrix formation. In particular, the study of the specific MC protease expression, tryptase, is of special significance. Using a combined histochemical approach, we have previously demonstrated increased tryptase expression in a mast cell population under various treatment protocols, primarily molecular hydrogen [21,22,23]. Tryptase has a high biological activity, affecting the state of many cellular and non-cellular components of the tissue microenvironment [14,24,25,26,27,28,29]. Concurrently, secreted MC proteases can lead to the further intensification of degranulation using an autocrine mechanism, and they can increase the liberalization of the biogenesis products in eosinophilic granulocytes [27,30]. Tryptase has its molecular targets on cells or components of the extracellular matrix, causing pro- or anti-inflammatory effects [27,31,32,33,34]. In general, tryptase most often initiates the development of inflammation, causing an increased permeability of the capillary wall, thus increasing the migration of neutrophils, eosinophils, basophils and monocytes outside the microvasculature [35,36]. These effects of tryptase may be mediated by the induced formation of kinins, IL-1 and IL-8 in the endothelium, which is combined with the altered synthesis of the intercellular adhesion protein ICAM-1. A number of studies have shown tryptase’s close involvement in the processes of angiogenesis. Tryptase, with the help of matrix metalloproteinases (MMP) activation, has the potential to provide far-reaching rearrangements of the extracellular matrix associated with the degradation of the fibrous component and the components of the basic substance, including laminin, fibronectin, a number of proteoglycans, etc. [14,24,25]. The effects of tryptase in relation to the cells of fibroblastic differon are known; these effects cause their active movement and mitotic division, and the stimulation of the synthesis of collagen proteins. As a result, tryptase-promoted wound healing and may lead to fibrotic effects [15,24,37].

Tryptase has a high selectivity for PAR-2 receptors in various connective tissue cells, potentiating the development of inflammation [38]. It is possible to polarize macrophages into the M1 phenotype, enhancing pro-inflammatory signaling involving the FOXO1 pathway [39]. After surgical interventions, an increased presentation of PAR-2 on the soft tissue cells significantly aggravates the course of the postoperative period. Researchers have discussed various mechanisms by which tryptase influences the growth and differentiation of new blood vessels [29,40].

Polyanion secretion into the extracellular matrix, which can be a key factor in the process of fibrillogenesis, is critically significant [15]. MCs alter the parameters of the specific tissue microenvironment, thus initiating the polymerization of collagen molecules and other components into a fiber; this is the crucial aspect of the changing of the physicochemical parameters of the extracellular matrix, contributing to the onset of the fibrous structure’s formation. This is definitely associated with a higher frequency of detection of MCs with impregnated fibers after burning, regardless of therapy or types of observation.

Therefore, the treatment of burn wounds is accompanied by the altered functional activity of MCs. In fact, MCs can be considered objects for the targeted influence of various pharmacological agents in relation to the process of collagen fibrillogenesis. Notably, we should not forget about the excessive formation of the fibrous extracellular matrix component, which leads to certain changes in the dermis and, further on, certain possible external signs. However, whereas the primary preservation of vital functions is most often reported, it is essential to ensure the most active regenerative abilities of the elements of a specific tissue microenvironment from the very onset of the therapy. Further study of the involvement of MCs in extracellular matrix remodeling seems to be relevant in this regard.

## 4. Materials and Methods

The authors performed an experiment involving 36 mature white Wistar rats, weighed 165–210 g, at the Research Institute of Experimental Biology and Medicine, N.N. Burdenko Voronezh State Medical University (VSMU). The animals were kept under standard vivarium conditions, with a 14 h light regime and unlimited access to food and water. All manipulations with animals were carried out according to Order No. 708-n “On approval of the rules of laboratory practice” issued on 23 August 2010, Ministry of Health and Social Development, and approved by the local ethical committee, N.N. Burdenko Voronezh State Medical University (Minutes No. 6, issued on 17 November 2016). The analysis of stained micropreparations was conducted at the Research and Educational Resource Center (NRRC) of innovative technologies for immunophenotyping, digital spatial profiling and ultrastructural analysis, Peoples’ Friendship University of Russia, and the Research Institute of Experimental Biology and Medicine, N.N. Burdenko VSMU.

### 4.1. Experimental Design

The animals were divided into 4 groups: the control group (*n* = 9); the experimental group with burn wound simulation without therapeutic effects and spontaneous healing (*n* = 9); the experimental group with application of the ointment having high antibacterial activity and containing silver sulfadiazine as an active substance on the wound surface (*n* = 9); and the experimental group with application of water with a high content of molecular hydrogen as a regional effect—the water was irrigated on the wound surface (*n* = 9). The molecular hydrogen solution was prepared according to the manufacturer’s instructions on an Aquela blue setup using an Aquela 8.0 cartridge. The concentration of hydrogen in the solution used for wound irrigation reached 8.0 mg/L (measurements were made using a dissolved hydrogen analyzer MARK-501, with a hydrogen sensor DV-501 (these names are transliterations of the Russian words “MAPK-501” and “ДB-501”) (LLC Vzor, Nizhny Novgorod, Russia). The ointment was applied on the wound surface daily; the water was irrigated on the wound surface daily. An ointment was applied to the wound surface in the amount of 1 g, which corresponded to 10 mg of silver. Irrigation of the wound with water with a molecular hydrogen concentration of 8.0 mg/L was carried out in a total volume of 2 mL. The results were assessed at 3, 7 and 14 days into the recovery period. The animals were withdrawn from the experiment by overdosing AERRANE inhalation anesthesia (Baxter Helthcare, USA) at the Zoomed Minor Vet Optima station (China).

Thermal burns were simulated under inhalation anesthesia using a TEC-3 Zoomed Minor Vet Optima evaporator (China). A YIHUA 8858 BGA Soldering Rework Station Portable Hot Air Gun (China) was used to simulate a burn injury. The burn injury resulted from the 40 s exposure of an element heated up to 80 °C at a 10 mm distance over the surface of a prior-shaved skin area in the interscapular region of the animals.

### 4.2. Histoprocessing

For morphological examination, a skin area with a burn surface and adjacent tissues were excised. The specimens were fixed in a neutral solution of 10% formalin for 48–72 h. The fixed material was embedded in a paraffin medium using an MTP SLEE histological processor (Germany). Sections 5 μm thick were prepared from paraffin blocks on an Accu-Cut SRM 200 rotary microtome (Japan) for histochemical analysis.

### 4.3. Tissue Probe Staining

Histological skin sections were impregnated with silver and stained with toluidine blue [41] (Table 2). Combined histochemical techniques in which staining with Giemsa solution or toluidine blue was combined with silver impregnation were used to simultaneously detect mast cells and the fibrous component [15].

### 4.4. Image Acquisition

Stained tissue sections were observed on a ZEISS Axio Imager.A2 equipped with a Zeiss alpha Plan-Apochromat objective 100x/1.46 Oil DIC M27, a Zeiss Objective Plan-Apochromat 150x/1.35 Glyc DIC Corr M27 and a ZEISS Axiocam 712 color digital microscope camera. Captured images were processed with the software program «Zen 3.0 Light Microscopy Software Package», «ZEN Module Bundle Intellesis & Analysis for Light Microscopy», «ZEN Module Z Stack Hardware» (Carl Zeiss Vision, Germany) and submitted with the final revision of the manuscript at 300 DPI. The number of collagen fibers with different morphometric characteristics in the pericellular region of the mast cells was determined using open-source software for digital pathology image analysis QuPath [42], with further calculation of the relative content (Table 1).

### 4.5. Statistical Analysis

Statistical analysis was performed using the SPSS software package (Version 13.0). The results are presented as mean (M) ± m (standard error of the mean). To assess the significance of the differences between the two groups, the Student’s *t*-test or the Mann–Whitney U test in the case of a nonparametric distribution were used.

## 5. Conclusions

Collagen fibrillogenesis is an important sign of wound healing processes, which determines the efficiency of the regeneration of the extracellular matrix in an area of injury. In skin repair, MCs play an important role, and we have shown only one of the many points of application for their regulatory action, which can be influenced by molecular hydrogen. Based on the results of the conducted studies, we suggest that MCs are important targets of molecular hydrogen in the local tissue microenvironment. Molecular hydrogen, by changing the regulatory activity of MCs, indirectly affects the rate of development of the inflammatory reaction, the formation of a certain portrait of the immune landscape, the remodeling of the extracellular matrix, and the restoration of the structural components of the skin. The favorable effects of molecular hydrogen on the post-burn restoration of the integrity of the skin as an organ allows us to consider it as a tool that has a positive effect on the local treatment of wounds in clinical practice.

## Figures and Tables

**Figure 1 pharmaceuticals-16-00348-f001:**
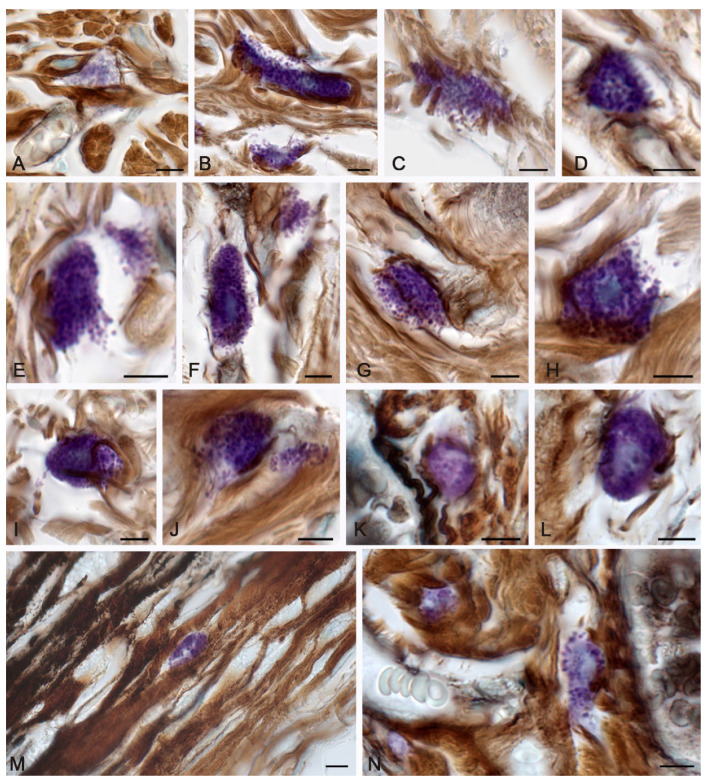
Mast cells of the skin dermis in the post-burn period. The group of spontaneous healing. (**A**–**J**)—in 3 days, (**K**–**N**)—in 7 days of the experiment. (**A**) A mast cell with a slight degree of metachromasia. (**B**,**C**) Active secretion of mast cells with granule localization in the peri-fiber space, with signs of the initial stages of fibrillogenesis (**C**). (**D**) Synchronous entry of secretory granules over a large area of MCs into the extracellular matrix. (**E**) Entry of metachromatic granules into the intercellular substance from different poles of the MCs, formation of a cytoplast with an active secretory potential. (**F**,**G**) Granule secretion caused inductive changes in the extracellular matrix, with formation of a fine network of fibers. (**H**,**I**) Removal of the granular material was not accompanied by fibrillogenesis. (**J**) Local secretion of MCs in the direction of collagen fibers, with formation of signs of the initial stages of fibrillogenesis. (**K**) A perivascular mast cell was involved in the initial stages of collagen fibers formation. (**L**) Active participation of MCs in fibrillogenesis. (**M**) Metachromatic secretory material is adjacent to the bundles of dermal collagen fibers of the skin burn surface. (**N**) Mast cells in the paracrine proximity of the vascular bed. Scale: 5 µm.

**Figure 2 pharmaceuticals-16-00348-f002:**
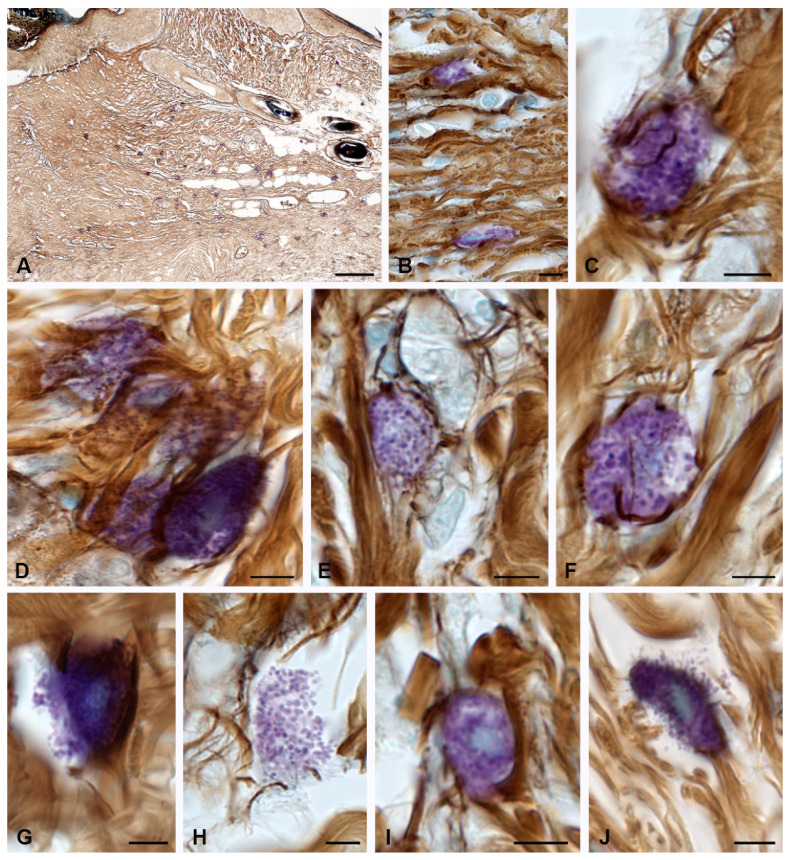
Collagen fibers of the skin dermis and mast cells 14 days after burning. (The group of spontaneous healing.) (**A**) Accumulation of MCs in a certain area of the skin dermis. (**B**) MCs in the regeneration zone under the burn surface of the skin. (**C**) Active fiber formation in the mast cell area. (**D**) MC colocalization in the remodeling cluster of the skin dermis. (**E**,**F**) Interaction of MCs and fibroblast (presumably) during collagen fiber formation. (**G**) Active secretion into the collagen fiber-free area. (**H**) A non-nuclear metachromatic fragment of the cytoplasm in the area of fibrillogenesis. (**I**,**J**) The initial (**I**) and final (**J**) stages of collagen fibers formation. Scale: (**A**)—100 µm, the rest—5 µm.

**Figure 3 pharmaceuticals-16-00348-f003:**
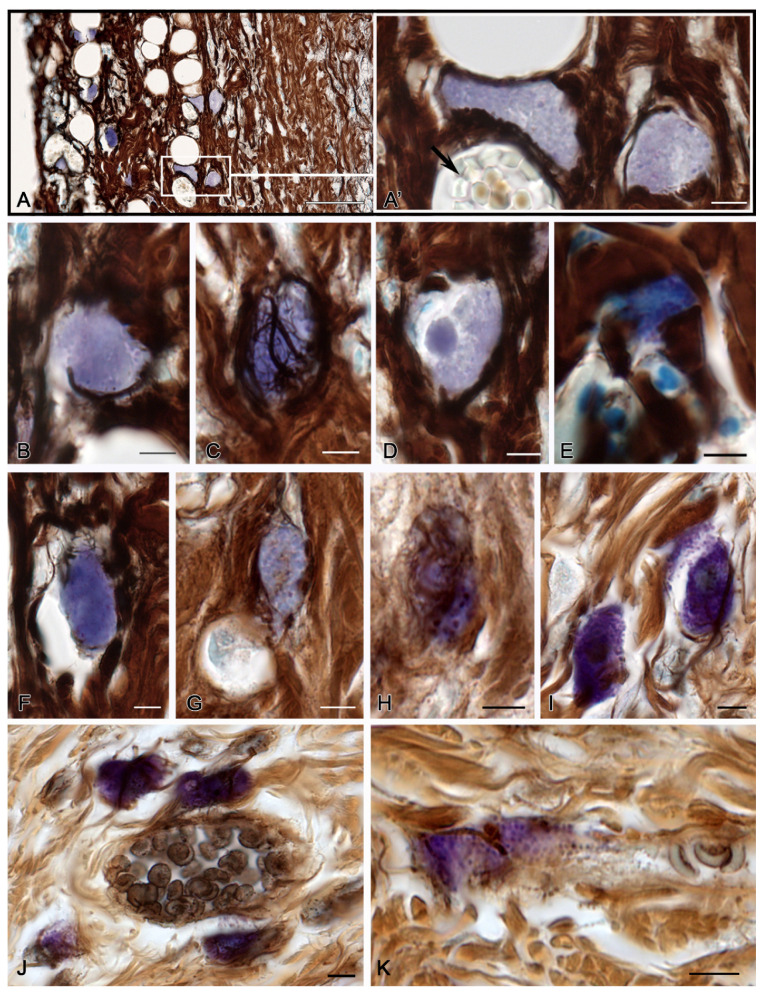
Mast cells of the skin dermis in the post-burn period under a therapeutic ointment application. (**A**–**I**)—in 3 days, (**J**–**K**)—in 7 days of the experiment. (**A**) Burned surface of the skin with a high content of MCs. (**A’**) An enlarged fragment of an (**A**) image. MCs are located next to the microvasculature (arrow). (**B**) MCs in the lacuna of the dermis after thermal exposure. Secretory granules are distinguished. (**C**) MCs in the burn surface, with a high level of fibrillogenesis. (**D**) MCs with altered localization of the secretome in the granules. The nucleus is not determined; a homogeneous formation with preserved properties of metachromasia is detected in the cytoplasm. (**E**,**F**) Homogenization of secretory material in MCs in the damaged area of the dermis with weak signs of pericellular fiber formation. (**G**–**I**) MCs in intact areas of the dermis with different levels of fiber formation intensity. (**J**) MC group in the peri-venular space of a specific tissue microenvironment, with signs of fibrillogenesis and active degranulation towards the basement membrane of the endothelium. (**K**) Degranulation of two MCs with the formation of an inductive zone of fibrillogenesis in the local area of secretion. Scale: (**A**)—50 µm, the rest—5 µm.

**Figure 4 pharmaceuticals-16-00348-f004:**
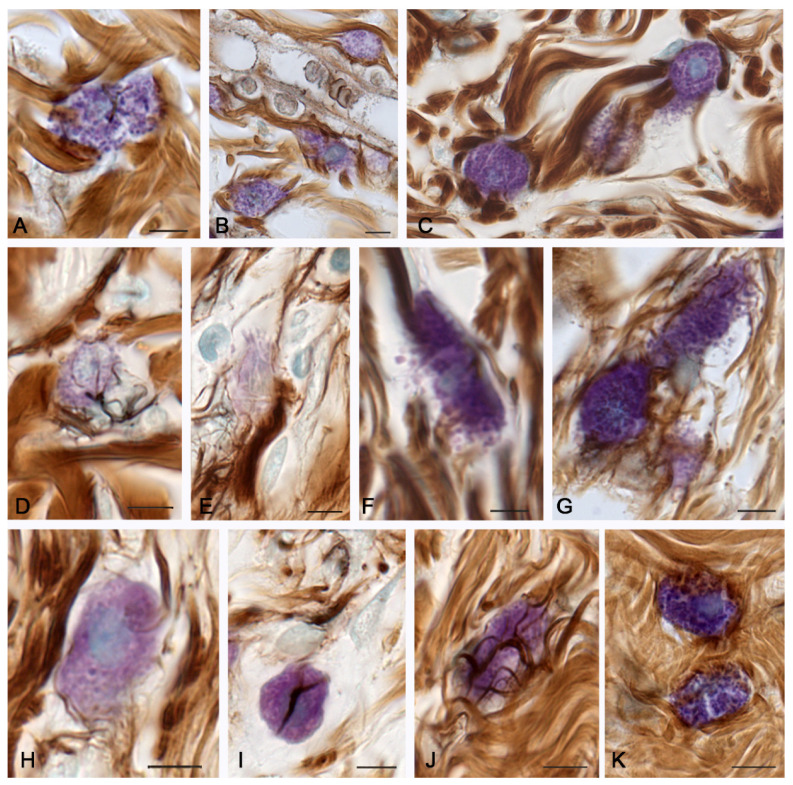
Mast cells of the skin dermis 14 days after the onset of the experiment under the therapeutic ointment application. (**A**) Close MC integration with bundles of collagen fibers. (**B**) Active MC participation in the formation of the microvascular stroma in the sub-burn zone of the dermis. (**C**) Joint participation of three MCs in targeted remodeling of the fibrous extracellular matrix of the skin dermis. (**D**) Initial stages of fibrillogenesis in the peri-burn area of the skin dermis with close MC interaction with fibroblast. (**E**) Active fibrillogenesis in the region of the MC fragment with low metachromasia. (**F**,**G**) Active MC degranulation into the loci of collagen fiber formation induction. (**H**) The initial stages of the collagen fibrils and fibers formation around the MCs. (**I**) Low activity of collagen fiber assembly initiation in the area of interaction between MCs and fibroblasts. (**J**,**K**) MCs in the zone of dense distribution of the fibrous extracellular matrix with high (**J**) and low (**K**) levels of fibrillogenesis. Scale: 5 µm.

**Figure 5 pharmaceuticals-16-00348-f005:**
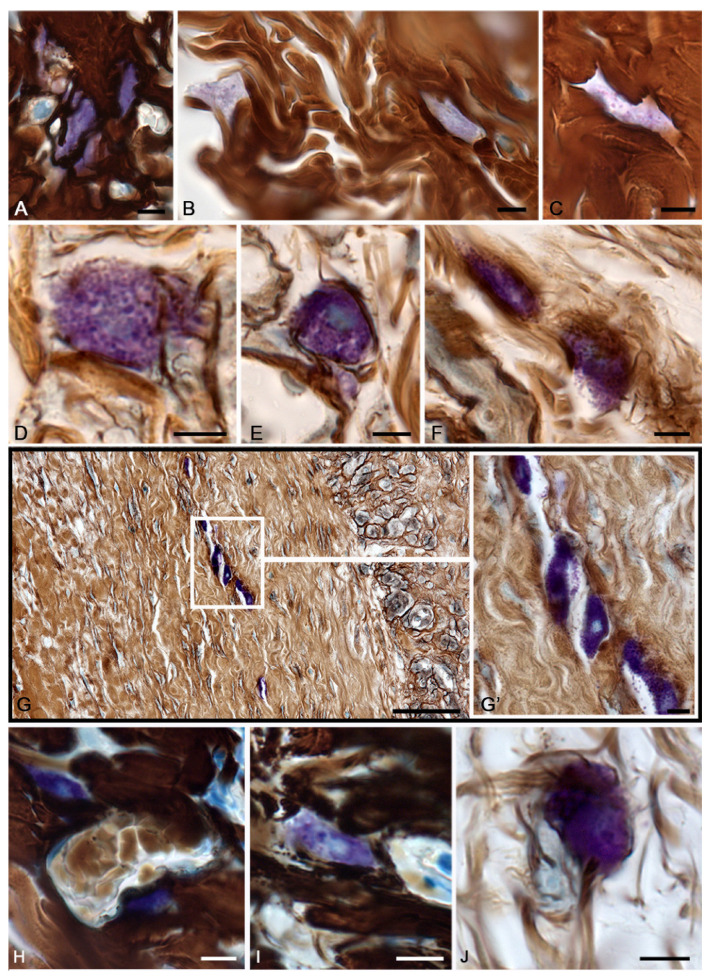
Mast cells of the skin dermis in the post-burn period under application of water enriched with molecular hydrogen. (**A**–**I**)—At 3 days, (**J**–**K**)—at 7 days of the experiment. (**A**–**C**) Different variants of MC localization in the damaged skin dermis. There is a homogenization of the secretome and a decrease in the properties of metachromasia (**C**). (**D**–**E**) MCs in the peri-burn skin dermis with formation of satellite fragments of the cytoplasm filled with secretome. (**F**) MC degranulation with collagen fibrillogenesis initiation. (**G**) Localization of the MC group in the area of the skin dermis restoration, (**G’**)—highlighted area at a higher magnification. The secretory activity of MCs is noticeable. (**H**–**I**) MCs in alternative areas of the skin dermis after a burn. Apparently, the MC secretome is partially preserved by this observation period. (**J**) Active participation of fibroblast-associated MCs in collagen fibrillogenesis. Scale: (**G**)—50 µm, others—5 µm.

**Figure 6 pharmaceuticals-16-00348-f006:**
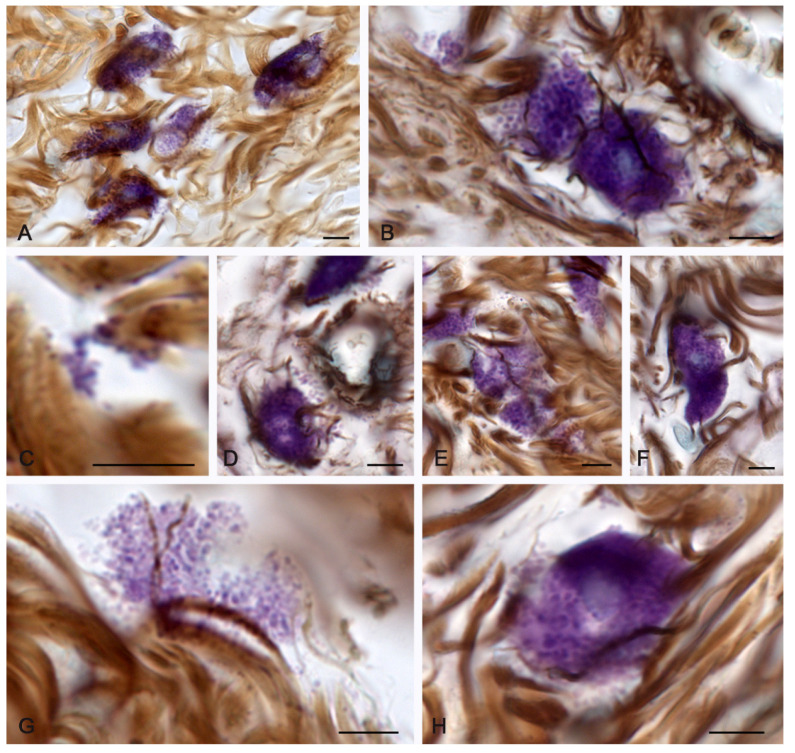
Mast cells of the skin dermis 14 days after the onset of the experiment under application of water enriched with molecular hydrogen. (**A**,**E**) Group of actively secreting MCs with the fibrillogenesis loci formation. (**B**) Two MCs with the reticular fiber formation at the site of adherence to each other. (**C**) Attachment of MC granules in the extracellular matrix to bundles of collagen fibers with the fibrillogenesis matrix locus formation. (**D**) Active secretion into the area of the vessel walls of the skin microvasculature. (**F**) Large MCs without signs of the reticular fiber initiation. (**G**) Nuclear fragment of the MC cytoplasm with the fibrillogenesis matrix formation. (**H**) Local MC secretion to the sites of the fibrillogenesis initiation. Scale: 5 µm.

**Table 1 pharmaceuticals-16-00348-t001:** Morphometric characteristics of fibrous structures associated with mast cells adjacent to the burn area of the skin dermis.

Experimental Groups		Relative Content of Fibers Adjacent to Mast Cells (in %, M ± m)			
		Absence of Impregnated Fibers in the Pericellular Zone	Fibrous Structures Diameter 0.2–0.5 µm	Fibrous Structures Diameter 0.5–1 µm	Fibrous Structures with a Diameter of More than 1 µm
The control group		24.2 ± 2.2 ^+,Δ^	6.4 ±0.8 ^+,∞,Δ^	28.2 ± 2.1 ^+∞ Δ^	41.2 ± 3.4 ^+,∞,Δ^
Day 3 of the recovery period	Spontaneous healing	32.7 ± 2.6% *^,∞,Δ^	22.7 ± 2.9 *^,∞,Δ^	13.2 ± 1.1 *^,∞,Δ^	31.4 ± 0.9 *^,∞,Δ^
	Application of ointment	24.4 ± 1.8 ^+,Δ^	57.4 ± 4.3 *^,+,Δ^	8.8 ± 0.9 *^,+,Δ^	9.4 ± 1.1 *^,+^
	Molecular hydrogen use	15.6 ± 1.1 *^,+,∞^	73.4 ± 5.2 *^,+,∞^	3.4 ± 0.5 *^,+,∞^	7.6 ± 0.9 *^,+^
The control group		24.2 ± 2.2 ^+,∞,Δ^	6.4 ± 0.8 ^+,∞,Δ^	28.2 ± 2.1 ^+,∞,Δ^	41.2 ± 3.4 ^+,∞,Δ^
Day 7 of the recovery period	Spontaneous healing	22.7 ± 1.9 ^∞^	52.7 ± 3.8 *^,∞^	8.2 ± 0.8 *^,∞^	16.4 ± 1.4 *
	Application of ointment	14.2 ± 0.8 *^,+^	68.4 ± 4.5 *^,+^	5.0 ± 4.4 *^,+^	12.4 ± 0.9 *^,+,Δ^
	Molecular hydrogen use	12.2 ± 1.1 *^,+^	67.4 ± 5.2 *^,+^	4.8 ± 0.4 *^,+^	15.6 ± 1.2 *^,∞^
The control group		24.2 ± 2.2 ^+,∞,Δ^	6.4 ± 0.8 ^+,∞,Δ^	28.2 ± 2.1 ^+,∞,Δ^	41.2 ± 3.4 ^+,∞,Δ^
Day 14 of the recovery period	Spontaneous healing	13.4 ± 1.1 *^,Δ^	55.5 ± 3.4 *^,Δ^	15.7 ± 1.1 *	15.4 ± 1.3 *^,Δ^
	Application of ointment	14.2 ± 0.9 *^,Δ^	54.7 ± 4.5 *^,Δ^	16.4 ± 1.2	14.7 ± 1.2 *^,+,Δ^
	Molecular hydrogen use	8.4 ± 0.7 *^,+,∞^	65.2 ± 4.1 *^,+,∞^	18.0 ± 1.4 *^,+^	8.4 ± 0.7 *^,+,∞^

Notes: * *p* < 0.05 compared to the control group; ^+^ *p* < 0.05 compared to spontaneous healing group; ^∞^ *p* < 0.05 compared to the ointment group; ^Δ^ *p* < 0.05 compared to the application of molecular hydrogen. The significance of differences was assessed at each observation period (3, 7 and 14 days).

**Table 2 pharmaceuticals-16-00348-t002:** Reagents used for histochemical staining of the rat skin.

Dyes	Catalogue Number	Provider	Dilution	Manufacturer
Toluidine blue	07-002	Biovitrum	Ready-to-use	ErgoProduction LLC, Russia
Giemsa solution, article number	20-043/L	Biovitrum	Ready-to-use	ErgoProduction LLC, Russia
Silver impregnation	21-026	Biovitrum	Ready-to-use	ErgoProduction LLC, Russia

## Data Availability

Data is contained within the article.

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
