# Peer review of "Mast Cells in Regeneration of the Skin in Burn Wound with Special Emphasis on Molecular Hydrogen Effect"

_pharmaceuticals, 2023, doi:10.3390/ph16030348_

Round 1
Reviewer 1 Report
I think that the topic is of interest, but I have some issues:
Why have they used an ointment like flammazine (silver sulfadiazine) and not others? What was ratio behind this decision? Many centers are not using this ointment anymore?
A lot of limitations are not cited and discussed in profound way. Please do so. What are the next steps? What is the real potential clinical impact. Please discuss!
Author Response
To Reviewer #1
The authors thank the Reviewer for the careful reading of the manuscript and for his valuable comments, which were correspondingly taken into account.
Comments and Suggestions for Authors
Reviewer: I think that the topic is of interest, but I have some issues:
Reviewer: 1.1. Why have they used an ointment like flammazine (silver sulfadiazine) and not others? What was ratio behind this decision? Many centers are not using this ointment anymore?
Authors: This work, first of all, was devoted to the evaluation of the effects of molecular hydrogen in the regeneration of a burn wound of the skin. The medicated ointment group was selected primarily for comparison with the effects of molecular hydrogen and the spontaneous healing group. Silver sulfadiazine, which is part of the ointment, is a broad-spectrum antimicrobial agent. The bactericidal properties are due to the activity of silver ions, which are released in the wound as a result of the dissociation of the silver salt of sulfadiazine; the release of silver ions occurs gradually (moderate dissociation), ensuring the constancy of the antimicrobial effect. The bactericidal activity of silver ions is complemented by the bacteriostatic effect of sulfadiazine (also released during the dissociation of the silver salt of sulfadiazine).
Reviewer: 1.2.A lot of limitations are not cited and discussed in profound way. Please do so. What are the next steps? What is the real potential clinical impact. Please discuss!
Authors: Thanks to the reviewer for the recommendation. The following text was included in the discussion:
Thus, collagen fibrillogenesis is an important sign of wound healing processes, which determines the efficiency of regeneration of the extracellular matrix in the area of injury. In skin repair, MCs play an important role, and we have shown only one of the many points of application of their regulatory action, which can be influenced by molecular hydrogen. Based on the results of the conducted studies, we suggest that MC are an important target of molecular hydrogen in the local tissue microenvironment. Molecular hydrogen, by changing the regulatory activity of MC, indirectly affects the rate of development of the inflammatory reaction, the formation of a certain portrait of the immune landscape, remodeling of the extracellular matrix and the restoration of the structural components of the skin. The favorable effects of molecular hydrogen on the post-burn restoration of the integrity of the skin as an organ allows us to consider it as a tool that has a positive effect in the local treatment of wounds in clinical practice.
Thank you
Igor Buchwalow

Reviewer 2 Report
The study report on the investigation of the role of mast cells in burn wound healing with and without treatment with a silver-based ointment or molecular hydrogen-containing water.
The aim of the study is very interesting and relevant for the understanding of wound healing processes and outcomes of therapies. The histological images of wound tissue with mast cell staining are very nice and well-presented and show a significant number of details that should demonstrate the role of mast cells at different stages of wound healing.
Nevertheless, this is not enough to state that there are differences between the different groups (treated vs untreated) or to demonstrate a specific mechanism of action. More measurements need to be done in order to obtain values that can be used for statistical analysis, e.g. ELISA detection of collagen types. The wound closure analysis by measuring wound surface over time is also an important measure that is missing. Finally, not enough information is given about the treatments: how much of the ointment and water were applied per wound? What were the applied amounts of silver and molecular hydrogen?
Further comments:
- The abstract could be improved by describing more clearly the goals of the study, the methods, the main results and the conclusions.
- Same for introduction: add what was the aim of the study.
- Line 70-71. Not clear. What was actively secreted? Do you mean the granula were actively secreted by mast cells?
- Line 113: change “pf” with “of”
- Line 130-131 How can you be sure that the cells in the pictures are fibroblasts?
- Line 173-174. Add arrows to indicate the microvasculature, i.e. the capillary.
- Line 221. To state: “the highest fibril-forming activity of MCs compared to other experimental groups”, it is necessary to have further measurements with a measurable numerical outcome and a statistical analysis.
Author Response
To Reviewer #2
The authors thank the Reviewer for the careful reading of the manuscript and for his valuable comments.
Comments and Suggestions for Authors
Reviewer: The study report on the investigation of the role of mast cells in burn wound healing with and without treatment with a silver-based ointment or molecular hydrogen-containing water.
The aim of the study is very interesting and relevant for the understanding of wound healing processes and outcomes of therapies. The histological images of wound tissue with mast cell staining are very nice and well-presented and show a significant number of details that should demonstrate the role of mast cells at different stages of wound healing.
Authors: The team of authors thanks the reviewer for a thorough analysis of the study and a positive assessment of our work, as well as valuable comments and questions that we answered below:
Reviewer: 2.1.Nevertheless, this is not enough to state that there are differences between the different groups (treated vs untreated) or to demonstrate a specific mechanism of action. More measurements need to be done in order to obtain values that can be used for statistical analysis, e.g. ELISA detection of collagen types.
Authors: We thank the reviewer for the important remark and included in the results of the work the analysis of the total number of fibrous structures around mast cells (per 1 cell, Table No. 1), although we planned to use it in another publication. The morphometric analysis technique was performed using the open source software for digital pathology image analysis QuPath (Bankhead P., et al., 2017) (indicated in the Materials and Methods section). The data obtained are presented in the Table No. 1, which has been added to the article.
Reviewer: 2.2. The wound closure analysis by measuring wound surface over time is also an important measure that is missing. Finally, not enough information is given about the treatments: how much of the ointment and water were applied per wound? What were the applied amounts of silver and molecular hydrogen?
Authors: The necessary information is included in the "Materials and Methods" section: “On the wound surface, an ointment was applied in the amount of 1 g, which corresponded to 10 mg of silver. Irrigation of the wound with water with a concentration of molecular hydrogen of 8.0 ppm was carried out in a total volume of 2 ml.
Further comments:
Reviewer: 2.3. The abstract could be improved by describing more clearly the goals of the study, the methods, the main results and the conclusions.
Authors: As recommended by the reviewer, the abstract has been substantially modified:
Reviewer: 2.4. Same for introduction: add what was the aim of the study.
Authors: Added:
The aim of the study was to evaluate the effectiveness of the use of molecular hydrogen on skin regeneration processes associated with the activity of collagen fibrillogenesis and the activity of mast cells in the treatment of skin burn injuries.
Reviewer: 2.5. Line 70-71. Not clear. What was actively secreted? Do you mean the granula were actively secreted by mast cells?
Authors: Предложение «MCs were properly detected either in the lower dermis, near the hypodermis, or in areas surrounding the wound surface along the periphery, where they were actively secreted (Figure 1)» was corrected:
MCs were properly detected either in the lower dermis, near the hypodermis, or in areas surrounding the wound surface along the periphery, where they actively exported granules and mediators to the extracellular matrix.
Reviewer: 2.6. Line 113: change “pf” with “of”
Authors: Corrected
Reviewer: 2.7. Line 130-131 How can you be sure that the cells in the pictures are fibroblasts?
Authors: We assume that these cells belong to fibroblasts according to external morphological features, including the structure of the nuclei. However, since it is difficult to phenotype fibroblast with reliable objectivity by visual signs (simultaneous immunohistochemical detection of several markers is necessary), we added the word “presumably” to the figure caption: “(E, F) Interaction of MCs and fibroblast (presumably) during collagen fibers formation ".
Reviewer: 2.8. Line 173-174. Add arrows to indicate the microvasculature, i.e. the capillary.
Authors: Corrected:
"MCs are located next to the microvasculature (indicated by arrow)."
Reviewer: 2.9.- Line 221. To state: “the highest fibril-forming activity of MCs compared to other experimental groups”, it is necessary to have further measurements with a measurable numerical outcome and a statistical analysis.
Authors: The data are presented in table 1.
Thank you.
Igor Buchwalow

Round 2
Reviewer 2 Report
The authors revised the manuscript as suggested and added semi-quantitative data to support the results shown as images. The abstract, introduction and conclusions were improved. In the Material and Methods part please add the information about the amount of silver applied per wound with the ointment and the volume of applied water containing molecular hydrogen. In my opinion, the manuscript can now be accepted for publication.
Author Response
According to the Reviewer's comments, we added the information about the amount of silver applied per wound with the ointment and the volume of applied water containing molecular hydrogen. (See the Materials and Methods).